# A New Green Composite Based on Plasticized Polylactic Acid Mixed with Date Palm Waste for Single-Use Plastics Applications

**DOI:** 10.3390/polym14030574

**Published:** 2022-01-31

**Authors:** Noran Mousa, Emmanuel Galiwango, Sabeera Haris, Ali H. Al-Marzouqi, Basim Abu-Jdayil, Yousuf L. Caires

**Affiliations:** 1Chemical and Petroleum Engineering Department, United Arab Emirates University, Al-Ain 15551, United Arab Emirates; 201450098@uaeu.ac.ae (N.M.); emmag@uaeu.ac.ae (E.G.); Babujdayil@uaeu.ac.ae (B.A.-J.); 2Civil & Environmental Engineering Department, United Arab Emirates University, Al-Ain 15551, United Arab Emirates; 201790275@uaeu.ac.ae; 3Palmade Plastic Cutlery Manufacturing LLC, Dubai, United Arab Emirates; yousuf.caires@palmade.me

**Keywords:** biodegradable, PLA, PBAT, TEC, date palm waste, single-use plastics, green composites

## Abstract

Petroleum-based plastic is widely used in almost all fields. However, it causes serious threats to the environment owing to its non-biodegradable properties, which necessitates finding biodegradable alternatives. Here, date palm rachis (DPR) waste was used as a filler (30, 40, and 50 wt%) to form a biodegradable composite with polylactic acid (PLA) and achieve cost–performance balance. DPR–PLA composites were prepared using a melt-mixing extruder at 180 °C by varying mixing time, DPR composition, and plasticizer type and composition. The biodegradable testing specimens were prepared by compression molding and analyzed using physical, thermal, and mechanical characterizations. Scanning electron microscopy images indicated a uniform dispersion of DPR (90 μm) in the PLA matrix. The esterification reaction resulting from this interaction between DPR and PLA was confirmed by Fourier transform infrared spectroscopy. The 30 wt% DPR–PLA composite was considered the optimal composite with the lowest melt flow index (16 g/10 min). This work confirmed the superior effect of addition of 10 wt% of triethyl citrate (TEC) compared with polybutylene adipate terephthalate (PBAT) by the improvement in the elongation at break of the optimal composite from 2.10% to 4.20%. Moreover, the addition of 10 wt% of PBAT to the optimal composite resulted in a lower tensile strength (21.80 MPa) than that of the composite with 10 wt% of TEC (33.20 MPa). These results show the potential of using the proposed composite as an alternative material for single-use plastics such as cutlery.

## 1. Introduction

Food-grade petroleum-based plastics, such as polypropylene (PP), polyethylene terephthalate (PET), and polystyrene (PS), are used in almost all fields [1]. From a commercial point of view, these plastics are manufactured to be non0biodegradable to increase their lifetime, resistance to temperature, and durability. Non-biodegradable waste cannot be easily disposed of because it cannot be dissolved or decomposed by natural agents. In particular, such waste cannot be naturally broken down by organisms. Thus, these plastics act as a source of pollution in landfills and oceans and may remain in the environment for thousands of years without any degradation. These plastics are one of the main causes of air, water, and soil pollution [2]. Global plastic production increased from 1.5 million metric tons in 1950 to 368 million metric tons in 2019 [3]. These plastics are not eco-friendly and need to be replaced by biodegradable alternatives. Scientists have investigated several alternatives such as biodegradable plastics or incorporating certain biodegradable components into the plastics to enhance their degradability. The degradation of biodegradable material in landfills occurs through the enzymatic activity of microorganisms such as fungi and bacteria. The end products of the degradation process are CO_2_, water, CH_4_, and other natural substances [1].

Polylactic acid (PLA) is a good example of a biodegradable polymer with high mechanical properties and biocompatibility. PLA (C_3_H_4_O_2_)_n_ is produced by the fermentation of agricultural products such as corn, sugar cane, potato, and rice. PLA exhibits accepted properties for applications that involve food contact. PLA shows very good thermal processability when compared with other biodegradable biopolymers [2]. Some disadvantages of using PLA without filler could be the high cost of production and relatively low thermal stability. To optimize the cost-performance balance and enhance mechanical and thermal properties, biodegradable fillers such as lignocellulosic biomass could be a good solution instead of synthetic fibers such as glass. The advantages of biodegradable natural fillers are mainly their abundance, high stiffness, non-abrasiveness to the processing equipment, low density, and low cost [4]. 

The world contains about 120 million date palm trees. A typical date palm tree produces 20 kilograms of waste in the form of dry leaves per year. The UAE produces about 500,000 tons of date palm waste every year [5]. These are disposed of in landfills or burned in farms causing environmental pollution, which can lead to global warming and its consequences due to the huge amounts of CO_2_ produced from burning date palm waste. Date palm biomass is a lignocellulosic material made mainly from cellulose, hemicellulose, and lignin. However, cellulose is the most important skeletal component of wood carbohydrate, and the polysaccharide cellulose is an almost inexhaustible polymeric raw material with attractive properties and structure. Cellulose can be extracted from different parts of date palm waste such as leaves, rachis, and fiber. Hence, date palm waste can be used to reinforce different polymeric matrices such as polylactic acid.

This work utilizes UAE date palm waste for the production and preparation of green biodegradable composites intended for single-use plastics such as cutlery and packaging applications with competitive properties as shown in Figure 1. PLA was a suitable polymer that would be compatible with conventional biodegradable polymers to be used in the preparation of the date palm biomass–polymer composite. The possibility of utilizing whole date palm rachis (DPR) waste without extraction as a filler to produce DPR–PLA composite has not been studied before. This research was focused on the effect of the addition of biomass (30 wt%, 40 wt%, and 50 wt%) to the PLA matrix on the properties of the resulting composite material, including phase adhesion, degradation temperature, glass transition temperature, and melt flow index. The effect of a maximum of 10 wt% of two food-grade plasticizers (triethyl citrate (TEC) and polybutylene adipate terephthalate (PBAT)) in such composites to enhance the mechanical properties of the DPR–PLA composite material was studied for the first time. 

## 2. Materials and Methods

### 2.1. Materials and Date Palm Rachis (DPR) Preparations

PLA (Ingeo biopolymer 2003D for fresh food packaging; molecular weight = 181,744 g/mol; specific gravity = 1.24; Tm = 175 °C) was purchased from Nature Works LLC (Minnetonka, MN, USA). TEC was acquired from Sigma-Aldrich and used as a plasticizer. PBAT was obtained from Natur-Tec®, Chennai, India, and used as an alternative plasticizer to enhance the ductility of PLA.

The date palm waste used in this study was obtained from Al Foah Farm in Al Ain, which is managed by United Arab Emirates University (UAEU). The date palm waste was collected in the form of large pieces and categorized as fiber, rachis, and leaflet. The rachis parts, which contained the highest amount of cellulose compared with fiber and leaflets, were selected for this study. Rachis was washed to remove sand, dust, and other particles, followed by shredding to few-centimeter-long chips using a high-speed shredder (TEEBA, Dubai, UAE; Date Seed Grinding Machine). The shredded rachis was further chopped using a lopper and then dried in a laboratory oven at 80 °C for 24 h. The sample was then ground using an electric grinder (Yongkang Sufeng Industry and Trade Co., Ltd., Jinhua, China; electric grain grinder SUS304) (Figure 2) and sieved to 90-micron particles using an electric shaker with the corresponding sieve mesh (MATEST, Treviolo (BG), Italy; Sieve shaker motor operated A060-01).

### 2.2. Cellulose Extraction from DPR Waste

A weighed amount (10 g) of the sieved biomass from rachis was placed in an extraction thimble and inserted in the Soxhlet extraction apparatus. About 150 mL of benzene and ethanol in a 2:1 ratio was used as the extraction solvent over a 4- to 5-h period at the boiling point to remove any waxes, inorganic salts, proteins, and pectin using the method described by [6]. The dried sample after extraction in the form of a solid mixture containing cellulose, hemicellulose, and lignin was transferred to a 250 mL beaker. Then 200 mL of 0.1 M HCl was gradually added. The beaker was placed on a heater at a temperature of 100 °C for 2 h with continuous magnetic stirring. In the last stage, the dried filtrate in the form of a solid mixture containing lignin and cellulose was transferred to a 250 mL beaker. Then 200 mL of 0.1 M NaOH was gradually added and heated at 100 °C for 2 h with continuous magnetic stirring. After NaOH treatment, the solution was filtered. At the end of the filtration step, the solution consisted of lignin, keeping wet cellulose on the filter paper, which was transferred to a Petri dish to dry and used later as a filler in the formation of the biodegradable PLA-based composites.

### 2.3. Fabrication of Polylactic Acid (PLA)-Based Composites

A sieved DPR sample (90-microns) was added as a filler to the biodegradable polymer of PLA in three different compositions of 30%, 40%, and 50% by weight. The DPR was not subjected to any chemical treatments or surface modifications to avoid the use of chemicals in food-grade applications, reduce the production cost and save the energy used in cellulose extraction, whereas 30 wt% of extracted cellulose was used to reinforce PLA. Melt mixing was performed using a counter rotating two-screw extruder (MiniLab HAAKE Rheomex CTW5; Dreieich, Hessen, Germany) at a screw speed of 140 rpm. The control sample was prepared by blending pure PLA under the same conditions. Different extruder mixing trials for 30% DPR–PLA composite shows that the best composite color with a recovery of 79.9 wt% and output thickness of 0.15 cm was produced by the melting of PLA at 180 °C for 7 min along with plasticizers up to 10 wt% followed by the addition of DPR to be mixed for 3 min. However, the extruded bio-composite was re-melted via injection molding machine (HAAKE MiniJet Pro, Dreieich, Hessen, Germany) in which the homogenous molten plastic was injected at high-pressure of 500 bars to fill the 80 °C closed mold cavity of a particular shape in 15 s to produce lab test specimens such as dumbbell-shaped specimens is presented Figure 3.

### 2.4. Characterization Tests

#### 2.4.1. Thermogravimetric Analysis (TGA)

Thermogravimetric analysis was used to determine the degradation temperature of the DPR biomass, PLA and PLA-based composites in a thermogravimetric analyzer (Q-50, TA Instruments, New Castle, DE, USA). Under nitrogen atmosphere supplied as 100 mL/min, samples of 10–15 mg were heated from 20 °C to 800 °C at a heating rate of 10 °C/min. The thermal stability of the samples were calculated according to the ASTM E2550 method.

#### 2.4.2. Differential Scanning Calorimetry (DSC)

DSC (Discovery DSC 25, TA Instruments, New Castle, DE, USA) was used to determine the melting temperature of about 5 to 7 mg sample of PLA and PLA-based composites. The samples were heated up to 200 °C in an inert nitrogen atmosphere. However, the melting temperatures (T_m_), glass transition (T_g_), and cold crystallization temperature (T_cc_) were recorded from the second heating from 25 °C to 200 °C at 10 °C/min.

#### 2.4.3. Fourier Transform Infrared Spectroscopy (FTIR)

The FTIR analysis (ASTM E168, E1252) using an IRTracer-100 FTIR spectrophotometer (Shimadzu, Kyoto, Japan) was used for identifying samples by their ability to absorb infrared light at different frequencies to produce a unique “spectral fingerprint”. The DPR biomass, PLA, and PLA-based composites were scanned in the frequency range of 500 to 4000 cm^−1^ at a resolution of 4 cm^−1^ [7].

#### 2.4.4. Scanning Electron Microscopy (SEM)

Scanning electron microscope JSM-6390A (JEOL, Tokyo, Japan) was used to observe the cross section morphology of gold-coated DPR biomass, PLA, and prepared biodegradable composite samples. The SEM images of the filler particles in the polymer matrix were recorded at different magnifications and developed at an acceleration voltage of 5 kV.

#### 2.4.5. Tensile Strength Test

A MTS Model MH/20 universal testing machine from Shimadzu, Kyoto, Japan (Figure 4) was used to determine the tensile properties of the PLA and the optimal biodegradable composite after plasticization with load application velocities of 10 mm/s. 

#### 2.4.6. Melt Flow Index (MFI)

The change in melt flow index (MFI) of PLA with the addition of date palm rachis biomass waste was measured using the XRL-400 series Melt Flow Index equipment from Jinan Precision Testing Equipment Co., Ltd, Shandong, China. MFI is simply the mass flow rate expressed in grams per 10 min under a constant load of 2.16 kg according to ISO 1133 at the melting point of 180 °C. A weight of 5 g from the extruder output was broken into small pieces to be inserted in the barrel of the MFI machine. The re-melted bio-composite was ejected discontinuously in an approximately equal amount from the MFI die using an automatic cutter when the cycle time of 10 s was completed. Finally, the melted samples were measured using a digital balance, and the average was calculated in grams per 10 s. Then MFI of PLA and DPR–PLA composites was then calculated in grams per 10 min.

## 3. Results and Discussions

### 3.1. DPR–PLA Composites

#### 3.1.1. Thermogravimetric Analysis (TGA)

In order to avoid possible thermal degradation, the thermal characterization of any pure materials and their composites is a critical step before polymer processing. Figure 5 shows the derivative weight changes versus temperature curves of PLA and PLA-based composites and their corresponding weight losses. The first weight loss for DPR and the three bio-composites is between 60.0 °C and 100.0 °C, which corresponds to water vaporization from the wall structure, void space, and the interfacial bonding between biomass and PLA matrix [8]. The DPR exhibits two phases of degradation as presented in Figure 5a. The first phase has a weight loss of 27.4% at approximately 300.5 °C. This phase is classified by the decomposition of pectin, cellulose, and hemicellulose at the same time [9]. The second phase has an additional weight loss of 25.8% at around 353.9 °C, which can be explained by the decomposition of lignin. The extracted cellulose shows one phase of degradation at 306.1 °C, as shown in Figure 5b, thus confirming the agreement with the TGA findings regarding phase one of the DPR degradation. The DPR biomass indicates good thermal stability by showing the highest final remaining residue of about 23 wt% at 800.0 °C. On the other hand, the extracted cellulose shows less final remaining residue of about 4.8 wt% compared with untreated date palm rachis biomass.

Figure 5 shows that the weight loss of 5% of PLA occurs at about 338.9 °C while the DPR exhibits a 5% weight loss at about 206.3 °C. The extracted cellulose has a 5% weight loss at 57.0 °C, which is not preferable in extruder processing. This is due to the fast-partial pyrolysis and the undesired color change. Based on these results, the processing temperature was held below 200.0 °C to prevent the thermal degradation of DPR in the PLA matrix. PLA presents 75.4% weight loss at 379.3 °C and a final remaining residue of 0.3 wt%, which is in agreement with the results obtained by [8]. In addition, the thermal degradation process of pure PLA can be explained by two steps, beginning with dehydration followed by chain scission [10]. Moreover, the three bio-composites demonstrated an intermediate level of thermal stability that is noticeably lower than that of pure PLA and higher than the thermal stability of DPR. The increase of the biomass content in the PLA-composite matrix shifts the degradation temperatures of the bio-composites to lower regions towards the biomass degradation temperature. The bio-composites of PLA with a biomass content of (30 wt%, 40 wt% and 50 wt%) have degradation temperatures of 355.7 °C, 346.8 °C, and 336.0 °C, respectively, whereas their corresponding weight losses are 83.6%, 71.1%, and 52.4%, respectively, as shown in Figure 5a. The 30% cellulose-PLA composite shows a relatively lower degradation temperature of 312.3 °C with a lower weight loss of 60.9%, as presented in Figure 5b when compared to 30% DPR–PLA composite.

#### 3.1.2. Differential Scanning Calorimetry (DSC)

The DSC curves of PLA and PLA-based composites with DPR biomass content of 30 wt%, 40 wt%, and 50 wt% are presented in Figure 6, and their corresponding phase-transition data are recorded in Table 1. The heating range was chosen to be from 20.0 °C to 200.0 °C according to the TGA results that suggested the fabrication of the bio-composites below 200.0 °C to avoid thermal degradation. However, the second heating curves were recorded due to the existence of obvious thermal peaks. Figure 6 shows that the glass transition temperature of the pure polylactic acid (PLA2003D) is 59.1 °C which agrees with other researchers as they obtained it as 59.5 °C upon the second heating [11]. The glass transition temperatures of the PLA-based composites with a biomass content of 30 wt%, 40 wt%, and 50 wt% shifted to slightly lower temperatures of 56.4 °C, 55.5 °C, and 54.1 °C, respectively.

Also, the cold crystallization temperatures of the bio-composites reduced insignificantly compared to pure PLA. This reduction indicates the nucleation activity of the biomass as a filler [12]. Moreover, the decrease in T_cc_ is followed by a slight drop in the melting temperature (T_m_) of the PLA with the addition of 50 wt% biomass from 149.9 °C to 145.8 °C. The slight decrease in T_g_, T_cc,_ and T_m_ of the PLA upon the increase in biomass content is explained by the resultant weaker interactions. This can lead to an increase in mobility that promotes an increase in the degree of crystallinity [13]. Furthermore, Table 1 clearly displays that the range of melting enthalpy was equal to the cold crystallization enthalpy. This indicates that the DPR–PLA matrix was amorphous before heating in the DSC [14]. The 30% cellulose-PLA composite showed a quite similar T_g_, T_cc,_ and Tm but relatively higher ∆H_cc_, and ∆H_m_ when compared to 30% DPR–PLA composite as displayed in Figure 6 and Table 1.

#### 3.1.3. Fourier Transform Infrared Spectroscopy (FTIR)

To investigate the presence of potential interfacial interaction between PLA and DPR biomass, FTIR experiments were conducted as shown in Figure 7. The DPR exhibits typical vibration bands of various chemical functional groups that are found in cellulose, hemicellulose, and lignin. The vibration of the OH bond in the biomass spectrum is directly related to the broad and strong absorption band between 3200 and 3500 cm^−1^ as shown in Figure 7a. The peak at 2916 cm^−1^ is assigned to the vibration of the asymmetrical stretching of the CH bond in the biomass [15]. The absorption band at around 1735 cm^−1^ can be attributed to the carbonyl group (CO) and the ester group with stretching vibration in hemicellulose [16]. In addition, the lignin bands at 1247 cm^−1^ to 1400 cm^−1^ are indicative of OCH_3_ and CC bonds, respectively [17]. At 1424 cm^−1^, the absorbance is due to the existence of CH deformation in lignin, and the symmetrical bending of CH_2_ in cellulose [18]. However, the bending vibration of the CH group on the aromatic ring at 1371 cm^−1^ is assigned to the hemicellulose and cellulose [16].

The O–H peak in PLA is observed at 3295 cm^−1^ as presented in Figure 7b. The bands observed at 2916 cm^−1^ and 2848 cm^−1^ can be linked to the stretching of symmetric and asymmetric C–H in CH_3_. However, three main FTIR spectra regions can be observed for PLA and PLA-based composites. The first region at 1744 cm^−1^ is linked to the carbonyl (-C=O) stretching peak. The second region is assigned to the -CH–O- group (–C–O- stretching bond) in PLA at 1181 cm^−1^. The last region consists of three distinctive peaks, which are attributed to the stretching vibrations in the -O–C=O group (-C–O-) at 1128, 1082 and 1039 cm^−1^. The bands at 867 cm^−1^ and 753 cm^−1^ are related to the PLA amorphous and crystalline phases. These FTIR results are consistent with the findings of Popa and co-authors [19]. The most important changes in the bio-composites compared with pure PLA are found in the carbonyl (C=O) stretching vibration at 1744 cm^−1^. This peak slightly shifted towards low wavenumbers by the increase in the biomass content of the DPR–PLA composite. Cellulose possesses a higher number of -OH groups compared with other biomass components such as hemicellulose, pectin, and waxy substances. Due to this characteristic, the C=O bond in the PLA can develop cellulose hydrogen bonds by its carboxyl and terminal hydroxyl groups [20]. This interaction causes a sudden decrease in intensity of the C=O peak in 30% DPR–PLA composite [21]. However, the increase in the biomass content from 30 wt% to 50 wt% causes the gradual rise in the intensity of C=O stretching peak as shown in Figure 7b. This is because of the esterification reaction between –OH of the biomass, the carbonyl (C=O) and the terminal –COOH group in the PLA [22].

#### 3.1.4. Scanning Electron Microscopy (SEM)

The morphology, phase adhesion, and the distribution of the DPR filler in the PLA matrix were studied using SEM. Figure 8a shows the SEM micrographs of PLA resin, presenting brittle nature with continuous phase and smooth exterior with several streams. The rachis biomass was ground to 90 μm (Figure 8b). The DPR appears as aggregates of entities with some nanofibrils on the surfaces of the biomass particles. These observations could be an indication that the biomass particles are agglomerations of thousands of individual biomass nanofibrils. Analogous SEM images were observed for PLA and cellulosic fibers [23]. Uniform dispersion of the 30 wt% DPR filler with a particle size of 90 μm (Figure 8c) in the PLA matrix was observed. Another finding was that the biomass still existed as thin fibril bundle aggregates with good fiber orientation, and no real bundle separation occurred during the extrusion stage. The SEM micrographs of 40 wt% DPR–PLA (Figure 8d) and 50 wt% DPR–PLA (Figure 8e) show that the extent of PLA coverage is lower than that of 30% DPR–PLA composite. The addition of 40 wt% and 50 wt% biomass to the PLA develops a slightly rougher and coarser surface. This is because of the extra accumulation of the biomass in certain points and voids within the bio-composite and the insufficient quantity of PLA to offer real wettability when biomass is increased to 50 wt%. Furthermore, the observed voids on the SEM images may be due to the existence of water that might be formed during the fabrication process [24]. On the other hand, the SEM image in Figure 8g revealed relatively better adhesion of the extracted 30 wt% cellulose (90 μm) to the PLA matrix compared to the untreated 30 wt% DPR as shown in Figure 8c. 

#### 3.1.5. Melt Flow Index (MFI)

The changes in melt flow index (MFI) with the addition of date palm rachis is illustrated in Figure 9. The MFI of PLA was obtained to be 7.09 g per 10 min, which agrees with the reported values by Nature Works LLC and other researchers [25]. The addition of the biomass (30 wt%, 40 wt% and 50 wt%) to PLA causes a gradual increase of the MFI to 16, 31.57, and 54.78 g/10 min, respectively. This indicates the growth of extrusion throughput because a greater mass of the bio-composite flows through the extruder die at a particular time. A significant increase in the MFI from 14.15 to 68.60 g/10 min was observed by researchers in the PLA matrix with the addition of 30 wt% of ground chestnut shell [26]. On the other hand, the 30% cellulose-PLA composite showed an undesirable MFI of 79.15 g/10 min, which is five times that of the 30% DPR–PLA composite as presented in Figure 9. When the amount of biomass is increased in the PLA based composite, small particles of the biomass can penetrate and accumulate in the PLA matrix as displayed in the SEM images in Figure 8, thus developing easer slip and flow of the PLA matrix. The results of the melt flow index are well correlated with the previous DSC findings as well. DSC results showed a slight decrease in T_g_, T_cc_ and T_m_ of the PLA by the increase of the biomass content because of the resultant weaker interaction that leads to the increase in mobility. The 30% DPR–PLA composite showed lower MFI when compared to the other two PLA-based composites with 40 wt% and 50 wt% of DPR. For easier cutlery fabrication and processing in large-scale extruders and injection-molding machines, 30% DPR–PLA composite was selected and tested for cutlery purposes in one of the plastic production industries in China.

### 3.2. 30% DPR–PLA Composites with Plasticizers

#### 3.2.1. Thermogravimetric Analysis (TGA)

Thermogravimetric analysis was carried out on the plasticized 30% DPR–PLA composites to investigate the effect of the addition of triethyl citrate (TEC) and PBAT. Three levels of these plasticizers (1 wt%, 5 wt% and 10 wt%) were used, and the results are shown in Table 2. TGA analysis revealed that the maximum decomposition temperatures of all the plasticized composites are lower than both neat PLA and unplasticized 30% DPR–PLA composite. The degradation temperatures of the plasticized composites at maximum weight loss were shifted from 355.8 °C (30% DPR–PLA) to 336.7 °C and 332.9 °C with the addition of 10 wt% TEC and 10 wt% PBAT, respectively. All plasticized composites with TEC and PBAT showed good thermal stability, and their thermal degradation temperatures were higher than the controlled processing temperature of 180 °C.

#### 3.2.2. Differential Scanning Calorimetry (DSC)

The glass transition temperatures (T_g_) of the plasticized 30% DPR–PLA composites with TEC and PBAT were obtained from the second heating of the DSC curves and are presented in Figure 10. The T_g_ of the 30% DPR–PLA composite shows an insignificant reduction with the addition of 1 wt% PBAT. Then it remains constant with increasing PBAT content (Figure 10a). Similar observations were attained by Farsetti and co-authors [27]. The glass transition temperatures of the 30% DPR–PLA composite shift to lower temperatures with increasing TEC content, and it diminishes by the addition of 10 wt% TEC (Figure 10b). The reduction is due to the increase in the plasticizing effect of the low molecular weight TEC, which occupies intermolecular spaces and leads to poor interaction between PLA, DPR filler, and TEC. This will probably increase the polymer molecular mobility, which in turn reduces the glass transition temperatures [26]. Analogous reduction in glass transition temperatures was observed for PLA and chitin nanocrystal nanocomposites with increasing TEC content by some researchers [28].

#### 3.2.3. Scanning Electron Microscopy (SEM)

SEM analysis was used to observe the surface morphology of the plasticized 30% DPR–PLA composites fracture surfaces after tensile testing. SEM was used to investigate the effect of the addition of 1 wt% and 10 wt% of two plasticizers (TEC and PBAT) on the morphology and phase adhesion of the composite. Figure 11a shows the micrograph of PBAT resin, presenting ductile fracture with higher continuous smooth phase compared with neat PLA (Figure 11b). The addition of 1 wt% TEC (Figure 11d) shows slightly higher biomass coverage when compared to the plasticized 30% DPR–PLA composite with 1 wt% PBAT (Figure 11c). However, the incorporation of 10 wt% PBAT in the 30% DPR–PLA composite resulted in a smooth surface and exhibited well-dispersed biomass in the PLA matrix as shown in Figure 11e. Also, the presence of some PBAT of fewer than 5 μm in size was observed in the PLA matrix, which might be due to the transesterification reaction between PBAT and PLA. A high co-continuous phase and adhesion might be formed between the two phases at the higher composition of PBAT [26]. Moreover, the addition of 10 wt% of TEC in Figure 11f enhanced the dispersion of the biomass in the composite and reduced the large agglomeration of the biomass filler. A similar observation of well-dispersed chitin nanocrystals in the PLA matrix after the use of about 7.5 wt% TEC was reported [28].

#### 3.2.4. Tensile Strength

Mechanical tests were performed for PLA and 30% DPR–PLA composites to demonstrate the effect of the addition of three levels of two plasticizers on the mechanical properties of the composite material. Biodegradable plasticizers such as TEC and PBAT are utilized to enhance PLA ductility, flexibility, and processability. But this will be done at the expense of the reinforcing feasibility and tensile strength. Figure 12 shows that the tensile strength of pure polylactic acid (PLA 2003D) is 68.88 MPa, which agrees with other researchers’ findings [29]. The incorporation of 30 wt% of DPR filler clearly decreased the tensile strength of the PLA by 46%, as shown in Figure 12. This significant decrease can be explained by the development of biomass agglomerates, micro-voids, and porosity during the composite fabrication. Therefore, these developments in the PLA matrix will minimize the reinforcing feasibility of the DPR filler for carrying such applied stress. The tensile strength is improved from 31.82 MPa to 46.49 MPa by the addition of 1 wt% TEC to the 30% DPR–PLA composite during fabrication. A clear decrease by more than 10 MPa in tensile strength is observed by a further increase in the TEC composition up to 10 wt% as shown in Figure 12a. Moreover, the addition of 1 wt% PBAT to the 30% DPR–PLA composite as an alternative plasticizer to TEC increased the tensile strength from 31.82 MPa to 43.65 MPa as displayed in Figure 12b. A minimal decrease in the tensile strength to 39.06 MPa was noticed by the addition of 5 wt% PBAT. A significant reduction in tensile strength to 21.80 MPa was noticed by the incorporation of 10 wt% PBAT to the 30% DPR–PLA composite. This reduction was predicted due to the low tensile strength of the PBAT (21 MPa) compared to that of pure PLA [30]. The main food-grade petroleum-based products such as PET, PS and PP showed a close range of tensile strength, which varied from 25 MPa to 69 MPa [31,32].

#### 3.2.5. Elongation at Break

The elongation at break of PLA (2003D) as a rough and brittle polyester is 5%, as shown in Figure 13. This agrees with the research work of [33]. The addition of 30 wt% of DPR (90 μm) caused a significant decrease in the elongation at break to 1.80%. This is because of the absence of a total random uniform distribution of DPR in the PLA matrix, resulting in a higher tendency of the bio-composite to break. Other researchers have demonstrated a decline in the elongation at break as well with the increase in natural filler [34]. The 30% DPR–PLA composite needs to be plasticized to overcome rigidness and brittleness challenges. The results revealed that the 30% DPR–PLA composite’s ductility could be enhanced by incorporating just 1 wt% TEC, which improved the elongation at break from 1.80% to 3.23%. A minimal increase to 3.48% in elongation at break of the 30% DPR–PLA composite was observed by the addition of 5 wt% TEC. When 10 wt% TEC was used, the elongation increased to 4.20% to approach that of PET (5.2 ± 1.6)% [31].

On the other hand, the addition of 1 wt% PBAT showed relatively no effect regarding the elongation at break of the 30% DPR–PLA composite, as displayed in Figure 13. However, the elongation at break was enhanced to 2.25% after the addition of 5 wt% PBAT and remained the same even by increasing the plasticizer to 10 wt% PBAT. The resultant elongation at break is very close to that of PS (1–2.5%) [32]. Deng and co-researchers showed a significant increase in the elongation at break above 10 wt% PBAT [35]. The results for up to 10% plasticizer proved the superior effect of TEC compared with PBAT in terms of improving the elongation at break of the 30% DPR–PLA composite, as shown in Figure 13. This might be due to the penetration of TEC molecules within the interface between the PLA and the granules of biomass, thus reducing the binding force, and making it possible for molecular chains to move and slip, thereby enhancing the elongation [36].

## 4. Conclusions

In this study, a new green composite material was developed for single-use plastics such as cutlery and food-packaging applications. The composite was based on PLA mixed with rachis biomass at 30, 40, and 50 wt%. The biomass filler was obtained from DPR generated in the UAE. The DPR was used without any chemical treatments or surface modification to avoid the use of chemicals in food-grade applications, save energy, and reduce production cost. In addition, the composites of cellulose extracted from DPR, and PLA were prepared and compared with composites using the entire DPR as a filler. Melt mixing was performed at 180.0 °C, followed by the extrusion of the DPR–PLA composites, which were then remelted using an injection-molding machine to fill the closed mold cavity. The prepared DPR–PLA composites were investigated using different thermal, mechanical, and physical tests. TGA results revealed a weight loss of 5% in PLA at ~338.9 °C, while DPR demonstrated the same weight loss at approximately 206.3 °C. These values are crucial for preventing possible thermal degradation during processing. The Tg of PLA was 59.1 °C, while the Tg of the DPR–PLA composites shifted to a slightly lower temperature (54.1 °C) at a filler content of 50 wt%, which indicates an improvement in the ductility and processability of PLA.

FTIR results indicated a potential interfacial interaction between PLA and DPR. This interaction resulted in an esterification reaction of the –OH of DPR with the carbonyl (C=O), and the terminal–COOH group in PLA. Further, SEM results indicated that DPR appeared as aggregates with some nanofibrils on the surfaces of the DPR particles. The filler (30 wt%, particle size = 90 μm) was uniformly dispersed in the PLA matrix. Increasing the filler content to 40 and 50 wt% in the composite slightly increased the surface roughness and coarseness. Thus, the 30 wt% DPR–PLA composite was considered the composite with the optimal composition for processing in large-scale extruders owing to its lower MFI (16 g/10 min) compared with the other bio composites. The tensile strengths of pure PLA (2003D) and 30 wt% DPR–PLA composite were found to be 68.88 and 51.16 MPa, respectively. The incorporation of PBAT (10 wt%) showed much lower tensile strength (21.80 MPa) than that of the composite with 10 wt% of TEC (33.20 MPa). In addition, incorporating TEC (10 wt%) showed a significant increase in the elongation at break of the composite (from 2.10% to 4.20%).

In summary, this green composite can decrease the environmental burden caused by land and ocean pollution from petroleum-based plastics. The optimal plasticized 30 wt% DPR–PLA composite is suitable for single-use plastics such as cutlery application because it shows properties comparable with those of the petroleum-based plastics, especially in terms of tensile strength, elongation at break, and cost. Future investigation of the effect of food-grade biodegradable plasticizers and other additives (compatibilizer, anticaking agent, and antioxidant) on the proposed composite is recommended to further optimize the final composite characteristics in terms of water absorption, tensile strength, flexural strength, MFI, and cost. In addition, the determination of the crystallographic structure using X-ray diffraction analysis can give insights into regulating the mold-opening time to make the entire process more economical.

## Figures and Tables

**Figure 1 polymers-14-00574-f001:**
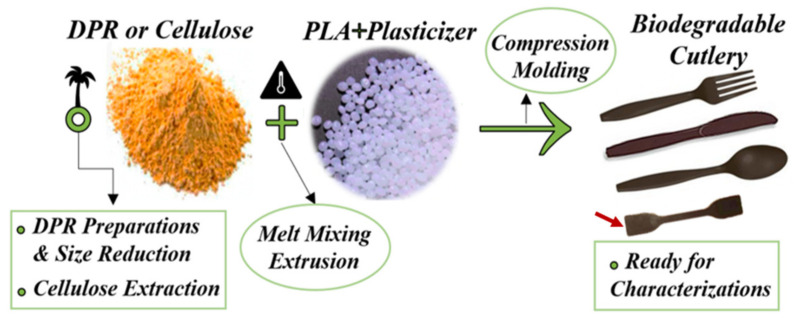
The fabrication methodology for the biodegradable date palm rachis–polylactic acid (DPR–PLA) composite.

**Figure 2 polymers-14-00574-f002:**
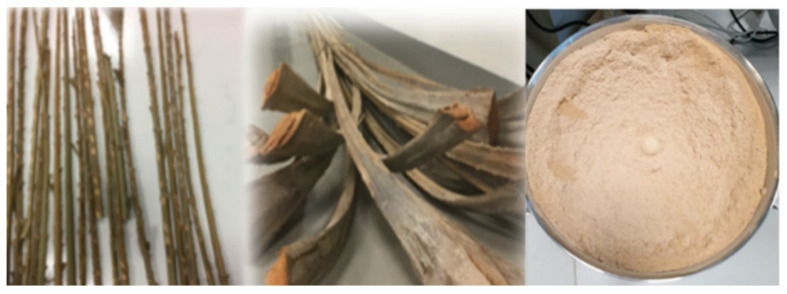
Date palm rachis preparations and size reduction.

**Figure 3 polymers-14-00574-f003:**
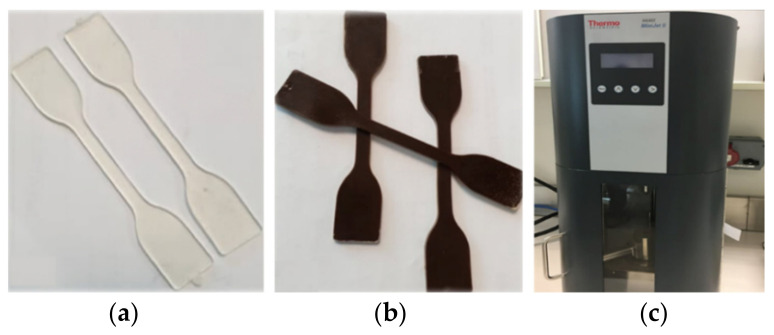
Dumbbell-shaped test specimens’ preparation. (**a**) PLA; (**b**) 30% DPR–PLA composite; (**c**) injection-molding machine.

**Figure 4 polymers-14-00574-f004:**
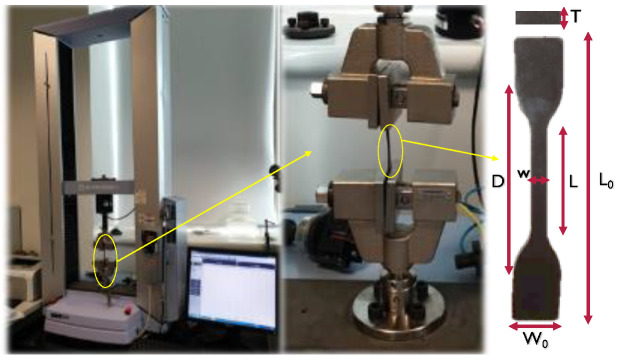
Tensile properties determination with the dumbbell-shaped specimens’ dimensions of T = 2 mm, w_0_ = 12 mm, w = 4 mm, L = 30 mm, D = 45 mm and L_0_ = 73.5 mm.

**Figure 5 polymers-14-00574-f005:**
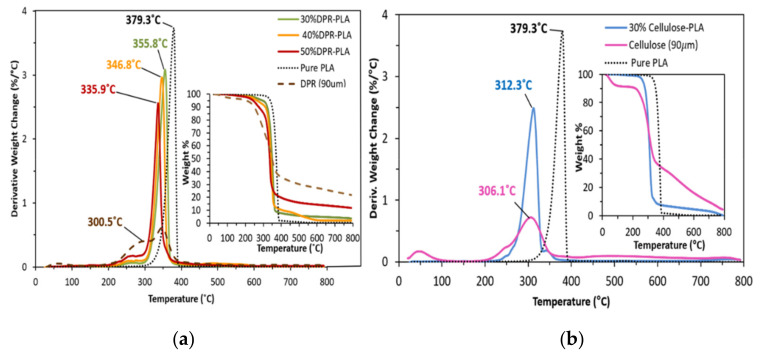
Derivative weight change and thermogravimetric analysis (TGA) plots. (**a**) DPR, PLA and DPR–PLA composites; (**b**) cellulose and 30% cellulose-PLA composite.

**Figure 6 polymers-14-00574-f006:**
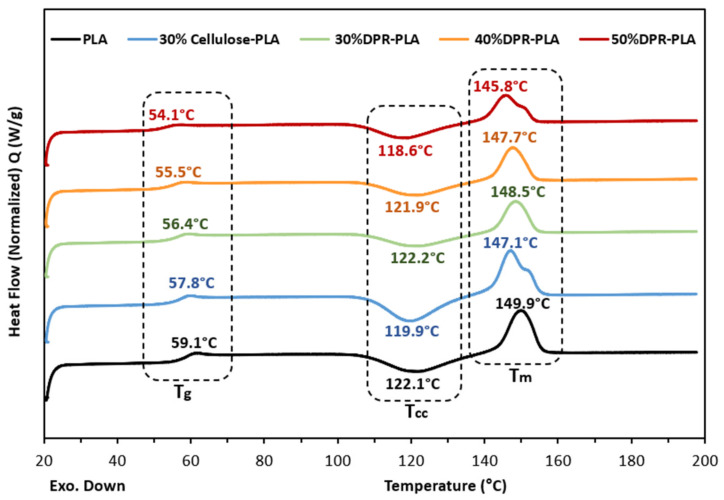
The differential scanning calorimetry (DSC) thermograms of PLA and PLA-based composites upon the second heating at 10 °C min^−1^.

**Figure 7 polymers-14-00574-f007:**
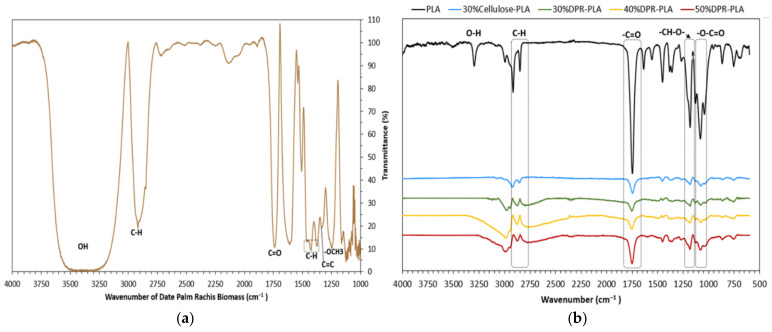
Fourier transform infrared (FTIR) spectra: (**a**) DPR; (**b**) PLA and PLA-based composites.

**Figure 8 polymers-14-00574-f008:**
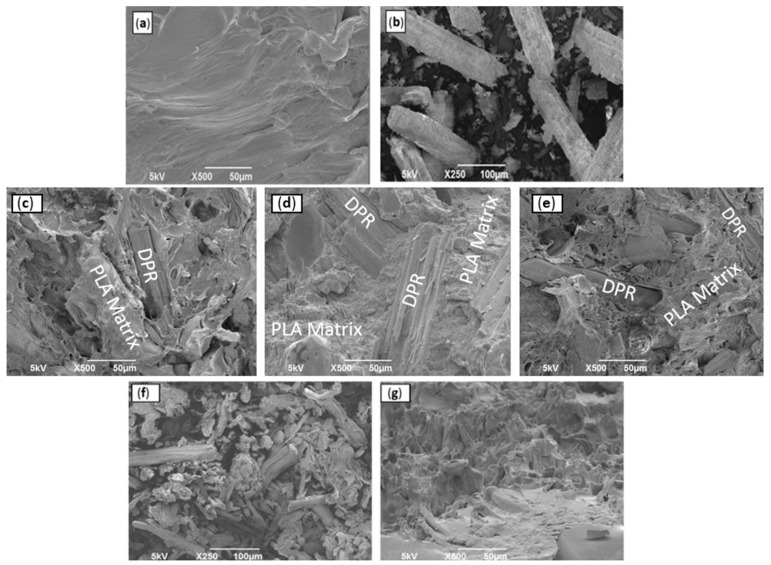
Scanning electron microscopy (SEM) images: (**a**) PLA2003D; (**b**) DPR (90 μm); (**c**) 30% DPR–PLA; (**d**) 40% DPR–PLA; (**e**) 50% DPR–PLA; (**f**) cellulose (90 μm); (**g**) 30% cellulose-PLA composite.

**Figure 9 polymers-14-00574-f009:**
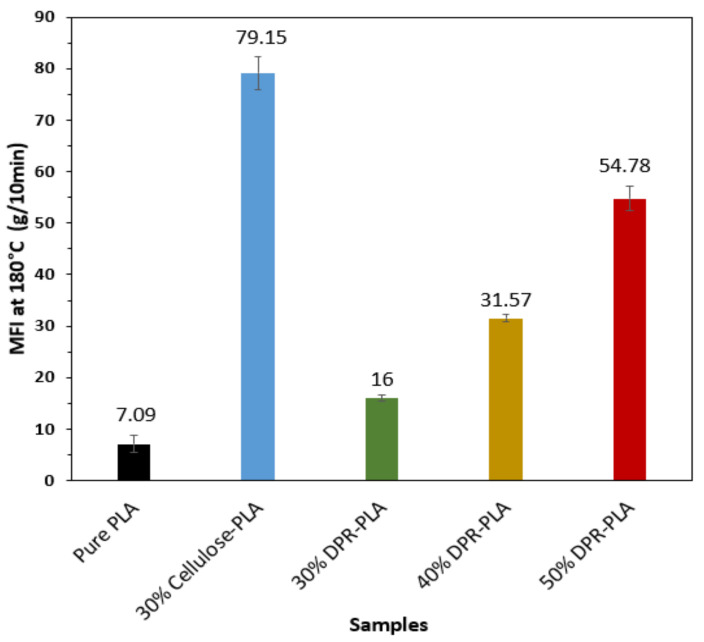
Melt flow index of PLA and PLA-based composites at 180 °C (g/10 min).

**Figure 10 polymers-14-00574-f010:**
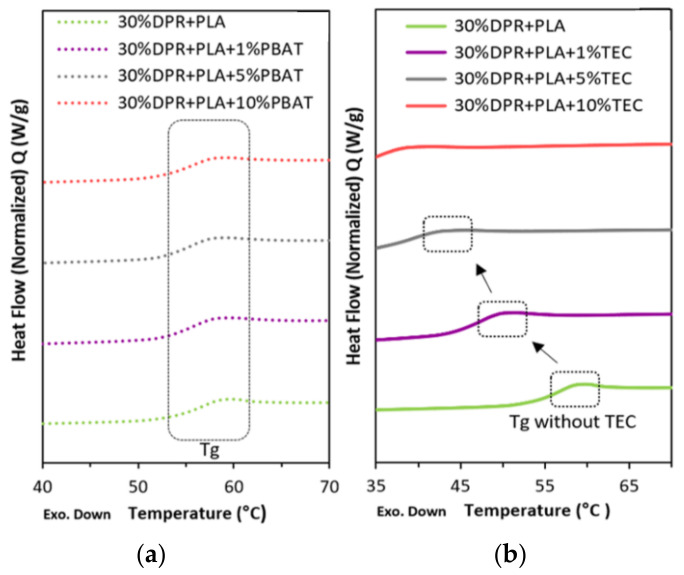
The glass transition temperatures (T_g_) from DSC thermograms of the plasticized 30% DPR–PLA with (**a**) PBAT and (**b**) TEC.

**Figure 11 polymers-14-00574-f011:**
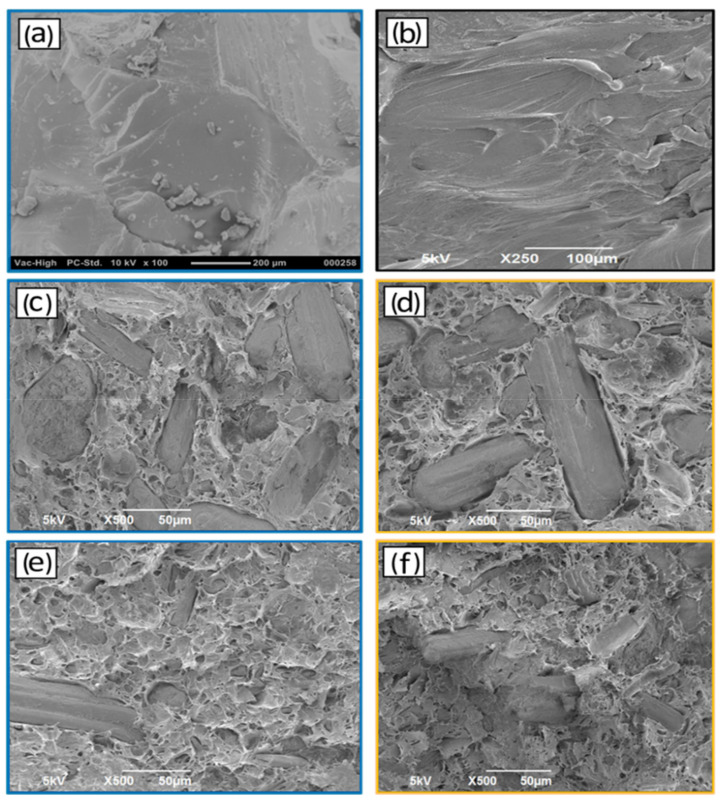
SEM images of (**a**) pure PBAT, (**b**) pure PLA, (**c**) 30% DPR + PLA + 1% PBAT, (**d**) 30% DPR + PLA + 1% TEC, (**e**) 30% DPR + PLA + 10% PBAT, (**f**) 30% DPR + PLA + 10% TEC.

**Figure 12 polymers-14-00574-f012:**
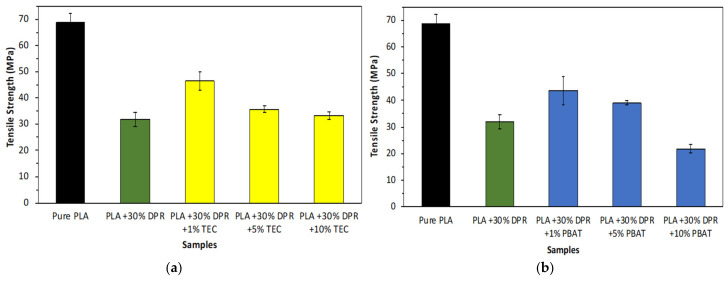
The tensile strength of 30% DPR–PLA composites with three levels of: (**a**) TEC; (**b**) PBAT.

**Figure 13 polymers-14-00574-f013:**
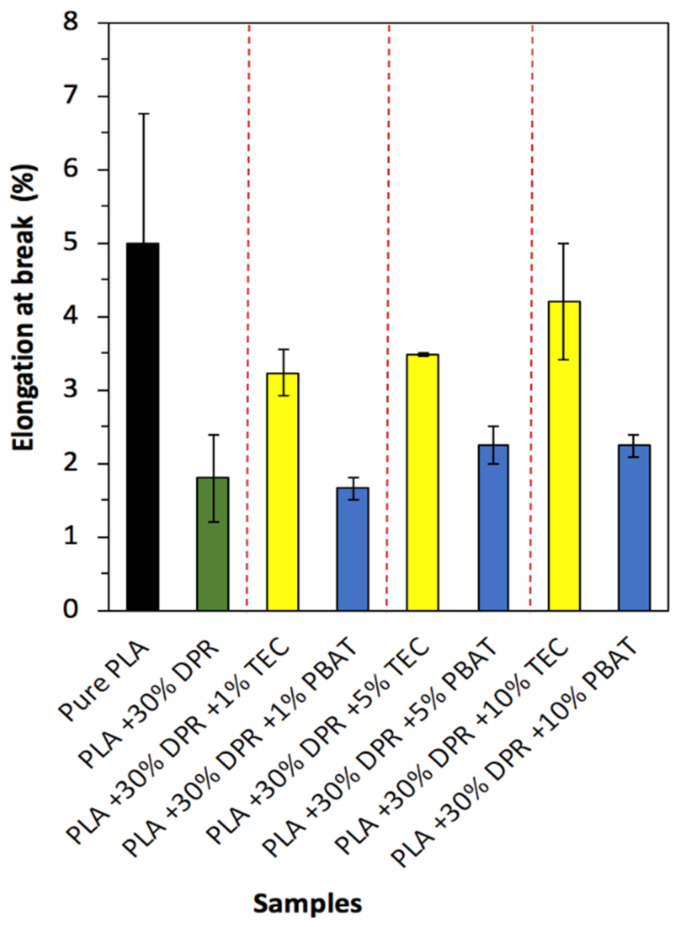
The elongation at break of PLA and its 30% DPR–PLA composites with three levels of two plasticizers (TEC and PBAT).

**Table 1 polymers-14-00574-t001:** The calorimetric data for PLA and PLA-based composites upon the second heating.

Sample	T_g_ (°C)	T_cc_ (°C)	T_m_ (°C)	∆H_cc_ (J/g)	∆H_m_ (J/g)
Pure PLA (polylactic acid)	59.1	122.1	149.9	21.2	22.9
30% Cellulose - PLA	57.8	119.9	147.1	23.6	23.9
30% DPR (date palm rachis) - PLA	56.4	122.2	148.5	14.3	15.00
40% DPR - PLA	55.5	121.9	147.7	14.2	17.1
50% DPR - PLA	54.1	118.6	145.8	14.1	14.7

**Table 2 polymers-14-00574-t002:** The calorimetric data for PLA and PLA-based composites upon the second heating.

Sample	T_5%_ (°C)	T_max_ (°C)	T_50%_ (°C)	Stability (%)
PLA ^a^	338.9	379.3	370.4	98.1
30% DPR ^b^ + PLA	291.9	355.8	347.8	93.8
30% DPR + PLA + 1% TEC ^c^	256.7	342.0	332.7	92.3
30% DPR + PLA + 5% TEC	207.6	333.0	324.6	85.7
30% DPR + PLA + 10% TEC	205.1	336.7	326.8	85.4
PBAT ^d^	364.9	403.9	400.0	99.0
30% DPR + PLA + 1% PBAT	243.3	330.3	324.4	92.1
30% DPR + PLA + 5% PBAT	246.9	330.3	324.4	92.1
30% DPR + PLA + 10% PBAT	252.8	332.9	328.4	92.6

^a^ polylactic acid, ^b^ date palm rachis, ^c^ triethyl citrate, ^d^ polybutylene adipate terephthalate.

## Data Availability

Data is contained within the article.

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
