# Peer review of "A New Green Composite Based on Plasticized Polylactic Acid Mixed with Date Palm Waste for Single-Use Plastics Applications"

_polymers, 2022, doi:10.3390/polym14030574_

Round 1

Reviewer 1 Report

The manuscript is on the development of a ‘green’ composite based on PLA and DPR (with or without plasticizers). It can be accepted after some modifications, as below:

  • The new composite may have potentials for food-packaging and disposable cutlery application. However, an assertion of new material for specific applications necessities the alignment of material's performances to the application requirements. The current manuscript does not cover that. The development of a composite without any chemical does not automatically qualify it for the mentioned applications. The paper does not provide results on its environmental / biodegradation either. Therefore, rather than mentioning the applications in the title and conclusion, the authors should limit the scope of the present paper to the development of green composite only.
  • Page 2, Line 80, the full form of TEC and PBAT should be provided in the paper, instead of in the abstract section only.
  • Page 3, Line: 121: What does it mean by “Sieved Date Palm Rachis (DPR) sample (53- and 90-microns)”. Are those the average diameter and the length of DPR or do they indicate something else? The authors should clarify that.
  • Page 3, Line 126: The details of the twin-screw extruder should be provided: is it a co- or counter-rotating?
  • Page 4, Line 157: Provide details of the sample preparation for the SEM. Are the specimens cryo-fractured/ coated with gold etc.? Are the micrographs viewed from the top down (surface) or sidewise (cross-section)?
  • Page 6, Line 203-232 and Table 1: Some of the temperature readings are provided up to the second and even third decimal places. What is the standard deviation/percent error of the measurement?
  • In general, authors should use the uniform reporting style for a specific set of the data (rounding of the decimal places up to appropriate digits), depending on the degree of precision of the experiment.
  • Page 7, Line 250: The wavenumbers in IR should be rounded to the whole numbers. The authors themselves mentioned the resolution to be 4 cm-1 in page 4, Line 156.
  • DRR has a fibrous shape. What do authors indicate by the size to be ‘90 microns’? From the visual estimation of SEM [Figure 8 (b)], the average length of DRR seems to be higher than 90 microns and diameter to be under 50 microns.
  • In Table-2, how was the % stability calculated?
  • Page-14, Line 470: “Further, SEM results indicated that DPR appeared as aggregates with some nanofibrils on the surfaces of the DPR particles”. That statement needs correction. The SEM results provided here do not indicate the presence of nanofibrils.

Reviewer 2 Report

Dear all,

Greetings

Please find enclosed my comments regarding paper

Referenced as: polymers-1574654

Titled: A New Green Composite Based on Plasticized Polylactic Acid Mixed with Date Palm Waste for Cutlery and Food Packaging Applications

 The authors have performed good and impressive article concerning the preparation of food tools such as cutlery or packaging using composite bio and natural waste products, but this paper can be accepted for publication in Polymers, after adressing and answering all these comments (Minor Revisions)

1) Title: very long

2) Abstract: please add the best condution of your composite to be a biodegradable

3) Keywords: ok

4) Comments:

  • your discuss in the introduction about plastics from petroleum please add the appropriete references concernin polypropylene [X], polyethylene tereppthalate[Y]...same for all your cited polymers
  • environmental issues lin air [R], water [P], and soil [Y], Please add references for each ones in all your introduction, insert references near to each properties
  • deplace title 2 (materials and methods) plus 2.1. materilas and DPR preparations in the page number 3
  • benzene used in the extraction is very toxic what about using toluene or xylene and ethanol?
  • do you have any idea about the price of your cutlery after using this composite (waste palm and PLA)
  • in the part 3.1.3. is better to draw chemical structure of your composite to facilate the interpretation of your IR spectrum
  • for figure 11 if you can add more interpretation about it

5) Conclusion: ok, please in your opinion which is the best combination of PLA pure or DPR pure or mixture for industrial purpose. If we gives you palm from other contrie it will have the same results?

6) References:

please update them add some of 2021 and 2022

With regards
